

# Comparative analysis and characterization of the gut microbiota of four farmed snakes from southern China

Bing Zhang[1], Jing Ren[1], Daode Yang[1], Shuoran Liu[1,2] and Xinguo Gong[3]

[1] Institute of Wildlife Conservation, Central South University of Forestry and Technology, Changsha, Hunan, China
[2] Institute of Eastern-Himalaya Biodiversity Research, Dali University, Dali, Yunnan, China
[3] Qiyang Gong Xinguo Breeding Co., Ltd, Yongzhou, Hunan, China

## ABSTRACT

**Background:** The gut microbiota plays an important role in host immunity and metabolic homeostasis. Although analyses of gut microbiotas have been used to assess host health and foster disease prevention and treatment, no comparative comprehensive study, assessing gut microbiotas among several species of farmed snake, is yet available. In this study, we characterized and compared the gut microbiotas of four species of farmed snakes (*Naja atra*, *Ptyas mucosa*, *Elaphe carinata*, and *Deinagkistrodon acutus*) using high-throughput sequencing of the 16S rDNA gene in southern China and tested whether there was a relationship between gut microbiotal composition and host species.
**Results:** A total of 629 operational taxonomic units across 22 samples were detected. The five most abundant phyla were Bacteroidetes, Proteobacteria, Firmicutes, Fusobacteria, and Actinobacteria, while the five most abundant genera were *Bacteroides*, *Cetobacterium*, *Clostridium*, *Plesiomonas*, and *Paeniclostridium*. This was the first report of the dominance of Fusobacteria and *Cetobacterium* in the snake gut. Our phylogenetic analysis recovered a relatively close relationship between Fusobacteria and Bacteroidetes. Alpha diversity analysis indicated that species richness and diversity were highest in the gut microbiota of *D. acutus* and lowest in that of *E. carinata*. Significant differences in alpha diversity were detected among the four farmed snake species. The gut microbiotas of conspecifics were more similar to each other than to those of heterospecifics.
**Conclusion:** This study provides the first comparative study of gut microbiotas among several species of farmed snakes, and provides valuable data for the management of farmed snakes. In farmed snakes, host species affected the species composition and diversity of the gut microbiota.

Corresponding author
Daode Yang, csfuyydd@126.com

## INTRODUCTION

Vertebrates have evolved intimate symbiotic relationships with their internal microbes, especially those that reside in the host gut (*Li et al., 2008*; *Gao, Wu & Wang, 2010*).

Studies of these symbiotic relationships have fundamentally increased our understanding of evolution, health, disease, and aging (*Kundu et al., 2017*). Gut microbiotas are extremely diverse, have unique functional characteristics, and may strongly affect the physiological functions of the host (*Costea et al., 2018*). For example, the gut microbiota may regulate the immune response, thereby affecting energy homeostasis (*Spiljar, Merkler & Trajkovski, 2017*) and nutrient metabolism (*Shibata, Kunisawa & Kiyono, 2017*). Changes in the gut microbiota may influence the functions of the brain and nerves (*Kundu et al., 2017*). Therefore, the gut microbiota may be an important factor determining the growth, immunity, and survival rate of farmed animals (*Hu et al., 2018*; *Rosshart et al., 2017*). The characterization of the gut microbiotas of farmed animals provides a scientific basis for disease diagnosis and health management (*Kohl, Skopec & Dearing, 2014*; *Jiang et al., 2017*; *Lyons et al., 2017*). Such characterizations are also essential for the commercial production of economically important animals and the conservation management of endangered species (*Larsen, Mohammed & Arias, 2014*).

Studies of gut microbiotas are primarily based on host fecal samples, as the collection of these samples is non-invasive. In mammals, fecal DNA reflects the composition and structure of the gut microbiota of the host (*Ley et al., 2008a*; *Costea et al., 2018*). Previous studies indicate that mammal gut microbiotas are dominated by Firmicutes and Bacteroidetes (*Ley et al., 2008a*; *Hu et al., 2017*). In birds, the microbiota demonstrates a similar phylum-level composition to that of mammals, being dominated by Bacteroidetes, Firmicutes, and Proteobacteria (*Waite & Taylor, 2014*). In reptiles, the gut microbiota also appeared to be dominated by Firmicutes, followed by Bacteroidetes and Proteobacteria (*Costello et al., 2010*; *Colston, Noonan & Jackson, 2015*; *Yuan et al., 2015*; *Jiang et al., 2017*). These results raise the possibility that there may be a certain phylogenetic relationship among gut microbiota of the amniotes (reptiles, birds, and mammals). A thorough characterization of the gut microbiota increases our understanding of gut microbial function, and, consequently, our ability to manipulate the gut microbiota to treat disease (*Kundu et al., 2017*; *Rosshart et al., 2017*; *Hu et al., 2018*). However, there have been few studies of the gut microbiotas of snakes, an ancient group with more than 3,000 extant species (*Uetz, Hošek & Hallermann, 2016*). Of the studies available, most investigated single species (*Costello et al., 2010*; *Colston, Noonan & Jackson, 2015*; *McLaughlin, Cochran & Dowd, 2015*; *Shi & Sun, 2017*). Therefore, it remains necessary to comparatively assess the composition, diversity, and phylogeny of snake gut microbiotas.

In recent years, several snake species have been successfully artificially bred on a large scale; such artificial-breeding programs not only satisfy commercial needs, but also reduce pressure on wild snake populations to some extent (*Hu et al., 2013*; *Hu, Tan & Yang, 2013*; *Li, 2009*). *Naja atra* (Elapidae), *Ptyas mucosa* (Colubridae), *Elaphe carinata* (Colubridae), and *Deinagkistrodon acutus* (Viperidae) are the snake species most commonly farmed in southern China (*Li, 2009*); *N. atra* and *P. mucosa* are listed in Appendix II of the Convention on International Trade in Endangered Species of Wild Fauna and Flora (1990; https://www.cites.org/).

The aim of this study was to characterize the fecal microbiotas of four different species of farmed snakes in southern China, and to evaluate the effect of host species on the

composition and diversity of the gut microbiota. This work serves as the first high-throughput sequencing analysis that compares the gut microbiotas of several farmed snake species. It is beneficial to study the gut microbiotas of snakes to improve the management of farmed snake populations.

# MATERIALS AND METHODS

## Sample collection

Fecal samples were collected from specimens of *N. atra*, *P. mucosa*, *E. carinata*, and *D. acutus*. All sampled snakes were healthy adults, hatched in 2014 and reared in similar farm environments. All snakes were kept in farming rooms with a temperature of 28 ± 2 °C, and a relative humidity of 80% ± 5%. Snakes were fed farmed chicks (*Gallus domestiaus*) and mice (*Mus musculus*). All snakes were fed once a week, all given the same food each feeding. For example, all snakes were fed chicks one time, and all snakes were fed mice the next time. The fecal matter of each snake was sampled after they were fed the chicks. Fecal samples from *N. atra*, *D. acutus*, and *P. mucosa* were collected at the Gong Xinguo snake farm, Yongzhou City, Hunan Province, China from July 8 to 11, 2017; fecal samples from *E. carinata* were collected at the Lvdongshan snake farm, Tujia-Miao Autonomous Prefecture of Xiangxi, Hunan Province, China on August 26, 2017. The wildlife operation licenses of the two snake farms were authorized by the Forestry Department of Hunan Province. The work was performed in accordance with the recommendations of the Institution of Animal Care and the Ethics Committee of Central South University of Forestry and Technology (approval number: CSUFT NS #20175167). The fecal sampling procedures used in this study were non-invasive to the snakes.

Individual snakes were farmed in plastic rearing boxes. The boxes were numbered to allow us to distinguish individuals. Individual snakes used for sampling were randomly selected. Fresh fecal samples from same individuals were collected using a sterilized sampling spoon and put in the same centrifuge tube: *N. atra* (group "Na"; $n = 6$), *P. mucosa* (group "Pmu"; $n = 4$), *E. carinata* (group "Ec"; $n = 6$), and *D. acutus* (group "Da"; $n = 6$). All fresh samples were immediately submerged in liquid nitrogen, and then frozen at −20 °C within 10 h. Samples were sent within 12 h on dry ice to the Wuhan Sample Center of Beijing Genomics Institute (BGI; Wuhan, China) for DNA extraction.

## DNA extraction, sequencing

Total DNA was extracted from the fecal samples using an E.Z.N.A. Stool DNA Kit (Omega Bio-tek, Inc., Norcross, GA, USA). The V4 hypervariable region of the 16S rDNA gene was amplified using polymerase chain reaction (PCR), with the primers 515F (5′-GTGCCAGCMGCCGCGGTAA-3′) and 806R (5′-GGACTACHVGGGTWTCTAAT-3′). PCR products were purified with AmpureXP beads (Agencourt; Beckman Coulter, Brea, CA, USA) to remove any non-specific amplicons. Qualified libraries were pair-end sequenced on a MiSeq System (Illumina, San Diego, CA, USA) with MiSeq reagents using the PE250 (PE251+8+8+251) sequencing strategy, following the manufacturer's

instructions. All libraries were sequenced on the Illumina MiSeq platform by the BGI (Wuhan, China).

## Bioinformatics and statistical analysis

The raw sequencing data were filtered, and the low quality reads were removed using an in-house procedure. The specific steps are as follows: (1) Sequence reads without an average quality of 20 over a 30 bp sliding window based on the phred algorithm were truncated, and trimmed reads with less than 75% of their original length and their paired reads were removed; (2) removal of reads contaminated by adapter (default parameter: 15 bases overlapped by reads and adapter with maximal three bases mismatch allowed); (3) removal of reads with ambiguous basa ($N$ base), and its paired reads; (4) removal of reads with low complexity (default: reads with 10 consecutive same base). The remaining high-quality reads were used for all subsequent analyses (*Fadrosh et al., 2014*). Paired end reads are merged to tags: If the two paired-end reads overlapped, the consensus sequence was generated by FLASH (Fast Length Adjustment of Short reads, v1.2.11), and the details of the method are as follows: (1) Minimal overlapping length: 15 bp; (2) Mismatching ratio of overlapped region: < = 0.1. Removal of paired end reads without overlaps (*Magoč & Salzberg, 2011*). Tags were aggregated into operational taxonomic units (OTUs) at 97% similarity using USEARCH v7.0.1090 (*Edgar, 2013*). Species annotation was then performed on the OTUs by comparing the OTUs to the 16S database (/RDP_set14/RDP_set14_NCBI_download_20151028) (*Cole et al., 2014*; *Quast et al., 2012*) with QIIME v1.80 package (confidence threshold: 0.60; *Caporaso et al., 2010*).

The bacterial species corresponding to the recovered OTUs were identified by comparing the OTUs to the species database (/RDP_set14/RDP_set14_NCBI_download_20151028). Profiling area maps and histograms for each sample set at the phylum, class, order, family, and genus levels were created. Heatmap analyses were also performed to compare bacterial community composition among the different host species. All bacterial classes with less than 0.5% relative abundance were combined into an "Others" class (*Henderson et al., 2015*; *Hu et al., 2017*; *Song et al., 2017*).

The representative sequences were aligned against the Silva core set (Silva_108_core_aligned_seqs) using PyNAST using "align_seqs.py." A representative OTU phylogenetic tree was constructed using the QIIME (v1.80) built-in scripts including the fasttree method for tree construction (*Caporaso et al., 2010*). The most abundant tags in each genus were chosen to represent the genus, and genus level phylogenetic tree was obtained by the same way of OTU phylogenetic tree. The phylogeny tree was imaged by software R (v3.1.1) (*R Core Team, 2014*) (*Caporaso et al., 2010*; *Costello et al., 2010*).

Within each sample, sequences were considered part of the same OTUs at a 97% similarity threshold. A Venn diagram was constructed based on these OTUs with the VennDiagram package (*Chen & Boutros, 2011*) in R (v3.1.1), showing the number of OTUs shared and unique among the different host species. A principal components analysis (PCA) was used to quantify the differences in OTUs composition among samples

and the distances between OTUs on a two-dimensional coordinate map. PCA was performed with the ade4 package (*Dray & Dufour, 2007*) in R (v3.1.1).

Alpha diversity describes species diversity at a single site or within a single sample (*Schloss et al., 2009*). Alpha diversity was estimated by calculating the observed species index and the Shannon index using mothur v1.31.2 (http://www.mothur.org/wiki/Calculators). Difference analysis and mapping were performed in R (v3.1.1) (*White, Nagarajan & Pop, 2009*). To compare differences in bacterial diversity between pairs of snake species, beta diversity was analyzed based on Bray–Curtis dissimilarity using QIIME v1.80 (*Caporaso et al., 2010*).

The cladogram and biomarkers images were generated using linear discriminant analysis effect size (LEfSe) (*Segata et al., 2011*). The one-sample Kolmogorov–Smirnov test was used to test the normality of the data. Then, we quantified the effect of host species on the five most abundant bacterial phyla using the general linear model (for the normally distributed data) or the generalized linear model (for the non-normally distributed data). A sequential Holm–Bonferroni correction was used to control for Type I error in SPSS v20.0 (IBM, Corp., Armonk, NY, USA). Differences in bacterial species abundance among samples were identified using the kruskal.test package (*White, Nagarajan & Pop, 2009*) in R (v3.1.1), adjusting for the false discovery rate and with the threshold $P$-value among groups set to 0.05. Based on these results, the bacterial species that most influenced the differences in sample composition among groups were identified.

### Availability of supporting data

The raw data obtained in this study have been deposited in National Centre for Biotechnology Information Sequence Read Achieve (Bioproject: PRJNA516815; accession numbers: SRR8494339–SRR8494360).

## RESULTS

### Data quality evaluation

Across all samples, 727,310 sequences with an average length of 252 bp were obtained (Table S1). The observed species and Shannon rarefaction curves tended to plateau, which showed that these sequence depths sufficiently captured the major microbiota in each sample (Fig. S1). Here, a total of 629 OTUs were obtained at the 97% sequence similarity cut-off levels and the number of OTUs shared by each sampling group was 109 (Table S1; Fig. S2). On average, 0.10% of all OTUs were unclassified at the phylum level (Fig. 1A), and 12.79% were unclassified at the genus level (Fig. 1B).

### Dominant bacterial taxa across all snake hosts

The gut microbiotas of the four farmed snake species fell into 15 phyla, 18 classes, 22 orders, 35 families, and 58 genera (Table 1; Fig. 1; Fig. S3). In the overall dataset, the five most abundant phyla were identified as Bacteroidetes (30.98%), Proteobacteria (24.80%), Firmicutes (20.96%), Fusobacteria (20.20%), and Actinobacteria (1.53%), while the five most abundant genera were *Bacteroides* (26.63%), *Cetobacterium* (19.06%),

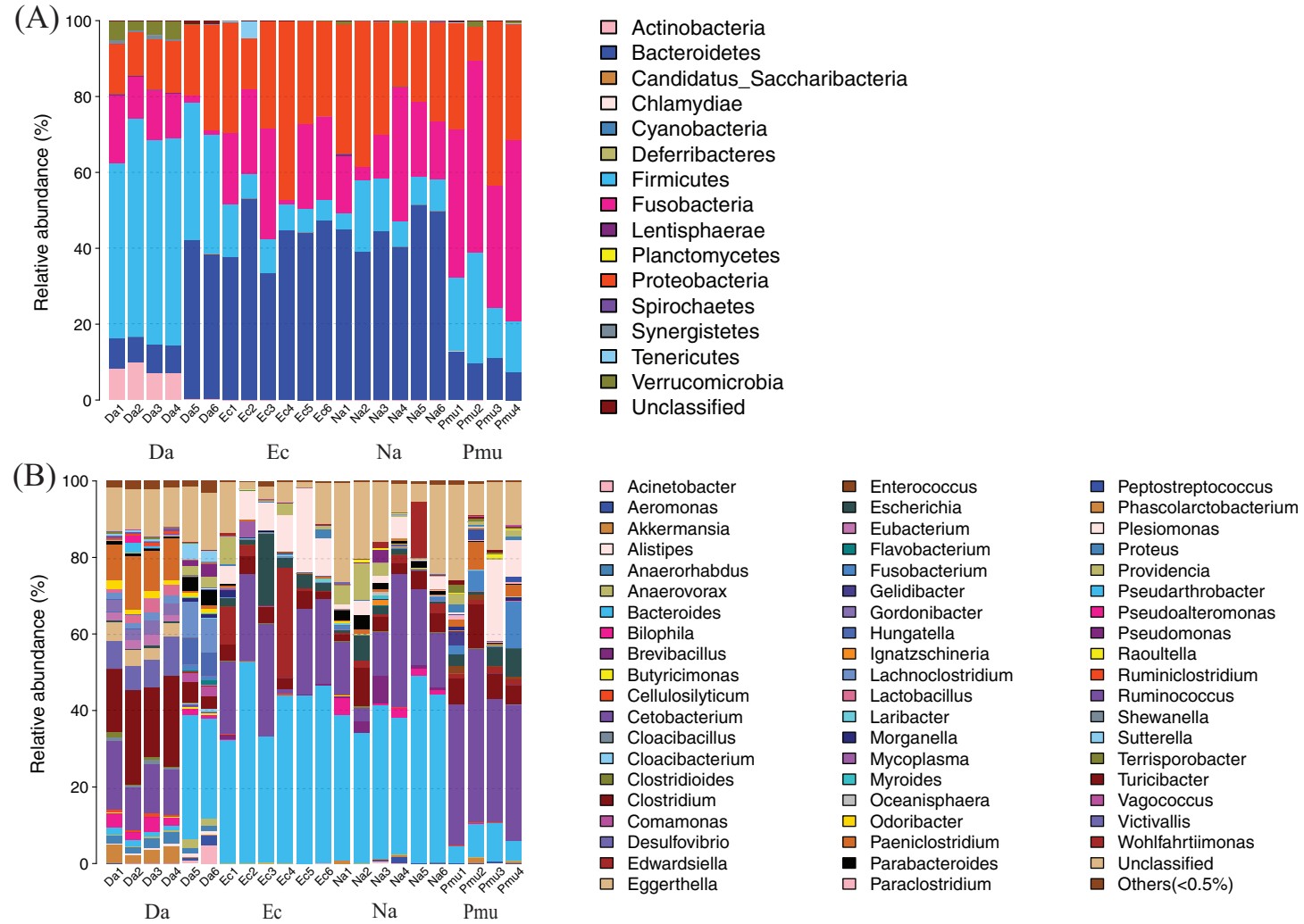

**Figure 1 Composition of the gut microbiotas of four snake species by bacterial (A) phylum and (B) genus.** Na, *Naja atra* group; Pmu, *Ptyas mucosus* group; Ec, *Elaphe carinata* group; Da, *Deinagkistrodon acutus* group.

*Clostridium* (7.84%), *Plesiomonas* (4.90%), and *Paeniclostridium* (2.89%) (Table S2). Phylogenetic analysis indicated that most genera fell into Bacteroidetes, Firmicutes, and Proteobacteria; only two genera fell into Fusobacteria (Fig. 2).

## Comparisons of gut microbiotas among the four snake species

### Alpha diversity analysis

Alpha diversity indices (observed species, $P = 0.001$; Shannon, $P = 0.002$) differed significantly among the four snake species (Figs. 3A and 3B). For the community richness estimator (observed species index), each pairwise comparison among three species (*D. acutus*, *E. carinata*, and *N. atra*) was significant, while *P. mucosa* was not significantly different from *E. carinata* or *N. atra*. For the community diversity estimator (the Shannon index), among three species (*D. acutus*, *E. carinata*, and *P. mucosa*) was significant, but *N. atra* was not significantly different from *E. carinata* or *P. mucosa* (Figs. 3A and 3B).

**Table 1 Composition of the fecal microbiotas of four snake species.**

| Group | Number of Phyla | Number of classes | Number of orders | Number of families | Number of genera |
|---|---|---|---|---|---|
| Na | 11 | 17 | 20 | 31 | 49 |
| Pmu | 11 | 16 | 19 | 28 | 44 |
| Ec | 9 | 15 | 19 | 27 | 44 |
| Da | 12 | 18 | 22 | 34 | 53 |
| Total | 15 | 18 | 22 | 35 | 58 |

Note:
Na, *Naja atra* group; Pmu, *Ptyas mucosus* group; Ec, *Elaphe carinata* group; Da, *Deinagkistrodon acutus* group.

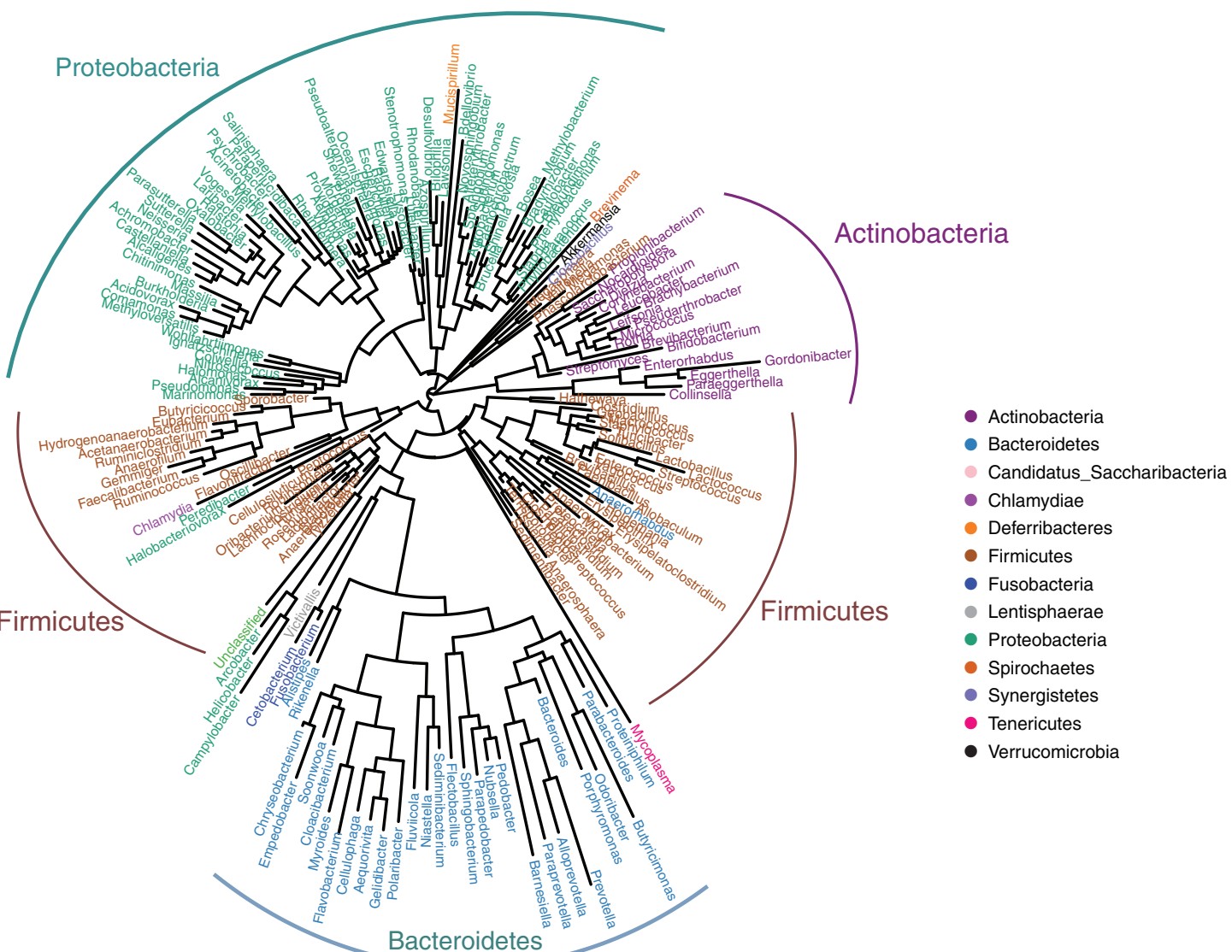

**Figure 2 Genus-level phylogeny of gut microbiota from four snake species.** Genera are colored by phylum.

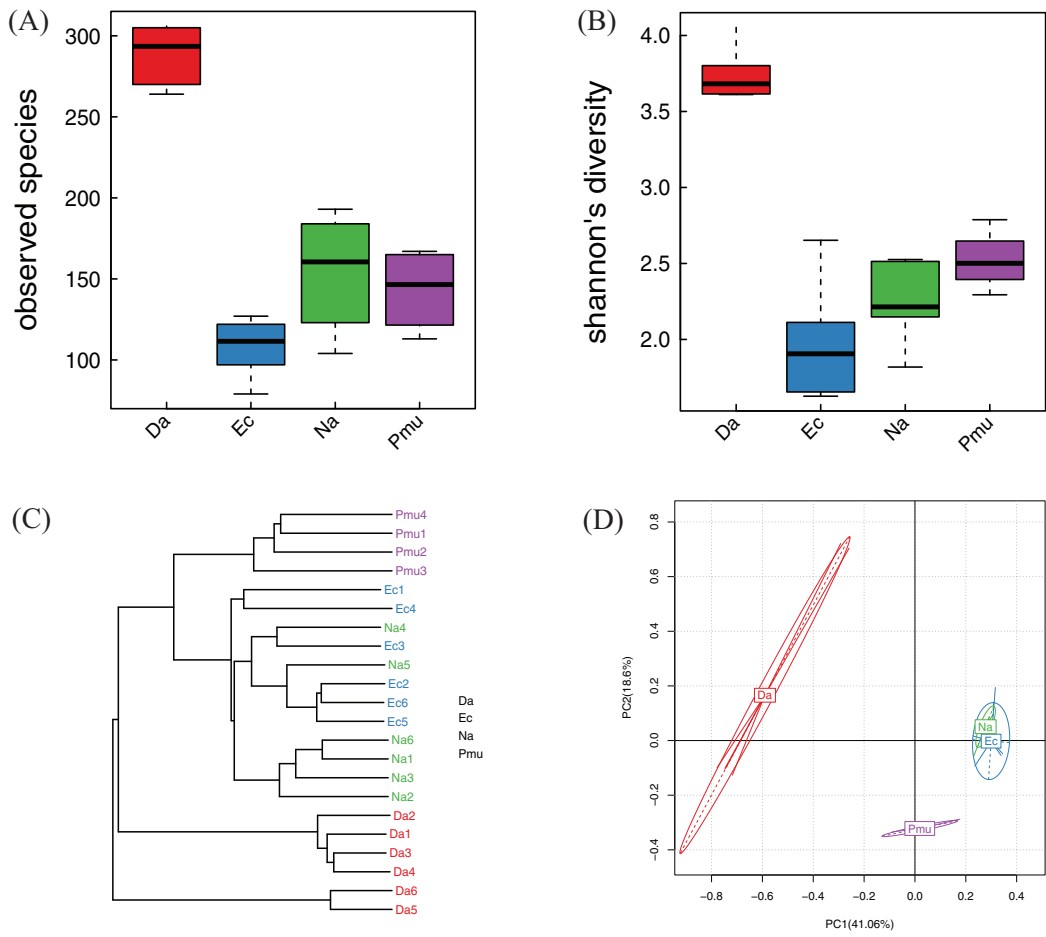

**Figure 3 Alpha diversity, beta diversity, and principal component analysis of the bacterial communities across four snake species.** (A) Observed species (Sobs) index and (B) Shannon's diversity index. The top and bottom of each box indicate the first and third quartiles, the line inside the box indicates the median, and the ends of the dotted lines represent the minimum and the maximum. (C) Cluster tree generated based on Bray–Curtis distances. (D) The variation explained by the plotted principal component is indicated by the axis labels. Na, *Naja atra* group; Pmu, *Ptyas mucosus* group; Ec, *Elaphe carinata* group; Da, *Deinagkistrodon acutus* group.

## Similarity analysis

The Bray–Curtis distance suggested that the bacterial community differences within each sample species were small; samples from the same species clustered together (with the exception of samples Na4 and Na5, which clustered with *E. carinata*; Fig. 3C). The PCA showed that the gut microbiotas from the same host species were more similar to each other than to the gut microbiotas from different host species, indicating that gut microbiotas were most similar within same snake species. Among the different snake species, *E. carinata* and *N. atra* were closest, indicating that the gut microbiotas of these two species were similar. In contrast, *D. acutus* was distantly separated from the other three species, indicating that the gut microbiota of *D. acutus* was dissimilar to those of the other three species (Fig. 3D).

Heatmap vertical clustering at the genus level showed that samples from the same snake species were tightly grouped on short branches, indicating that the composition and

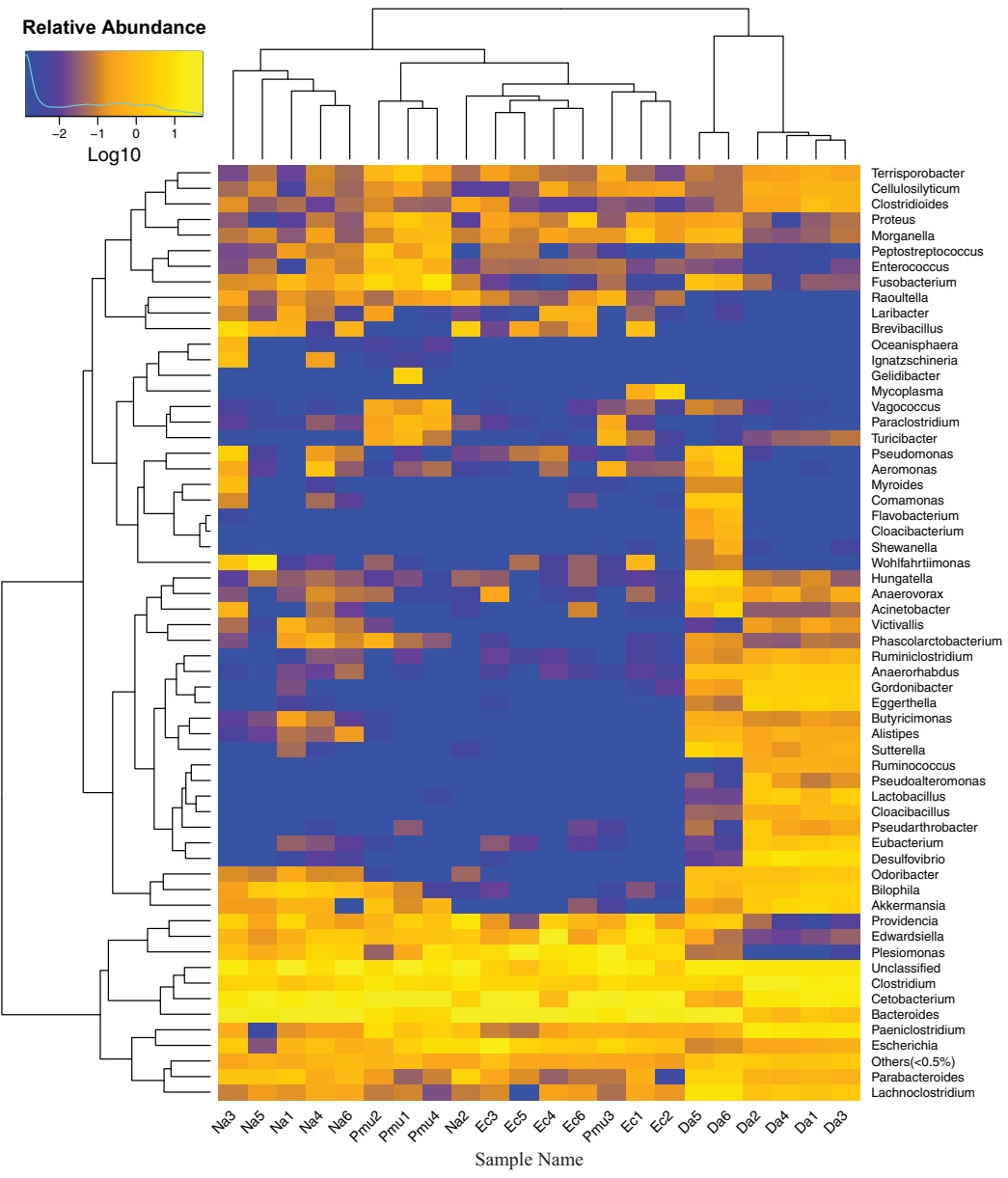

**Figure 4 Heatmap showing the genus-level bacterial community composition in the gut microbiotas of four snake species.** Na, *Naja atra* group; Pmu, *Ptyas mucosus* group; Ec, *Elaphe carinata* group; Da, *Deinagkistrodon acutus* group.

abundance of gut bacteria in the same sample were similar (with the exception of Na2 and Pmu3, which clustered with *E. carinata*; Fig. 4). These results were consistent with the beta diversity analysis.

## Differential microbes among species

The LEfSe analysis was used to screen the differential microbes among species. The cladogram also showed seven phyla, 11 classes, 17 orders, 29 families, and 45 genera were significantly enriched in distinct species (Fig. 5). The general linear model (GLM)

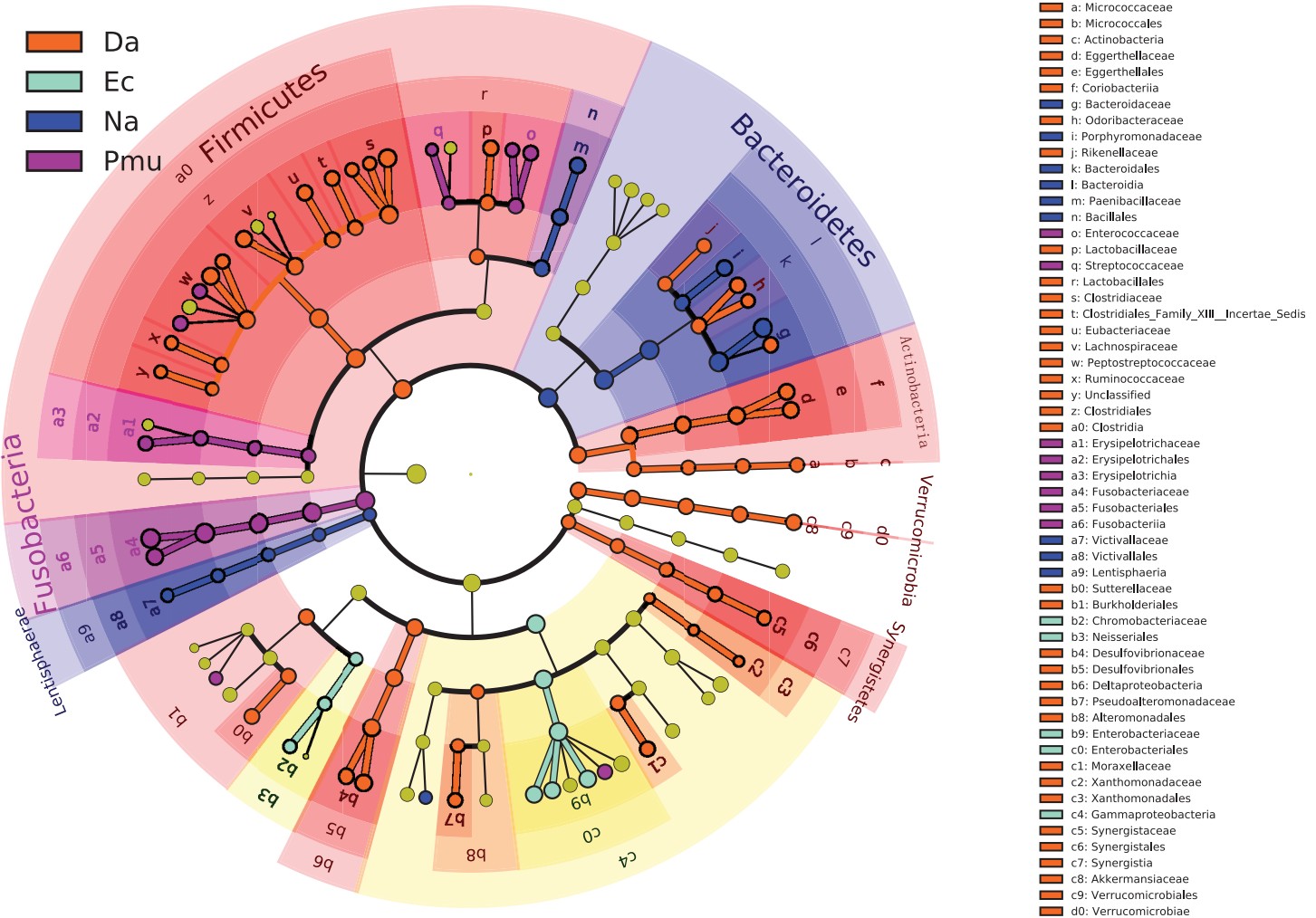

**Figure 5 A cladogram showing the differences in relative abundance of taxa at five levels across four snake species.** The plot was generated using the online LEfSe project. The orange, cyan, blue, and purple circles mean that four snake species showed differences in relative abundance, and yellow circles mean non-significant differences.

suggested that host species affected the relative abundances of Bacteroidetes, Firmicutes, and Fusobacteria (with the exception of Proteobacteria), whereas the GLMs identified no significant effects of species on Actinobacteria abundance (Table 2). The relative abundances of the five most abundant genera across the four host species was shown in Fig. S4. *D. acutus* had a significantly higher abundances of genera *Clostridium*, *Paeniclostridium*, and *Desulfovibrio*. *E. carinata* had higher abundance of genera *Edwardsiella*, *Escherichia*, and *Plesiomonas*. Compared with other species, *P. mucosa* showed greater significantly in the abundances of genera *Cetobacterium*.

## DISCUSSION

Tens of billions of bacterial species have colonized vertebrates, typically in the gut (*Ley et al., 2008b*; *Costea et al., 2018*). The composition and structure of the normal gut microbiota can be used to assess animal health and diagnose or prevent disease

**Table 2 The differences in relative abundance (% ± SD) of the top five most abundant phylum of four snake species.**

| Top five most abundant phyla | Na group | Pmu group | Ec group | Da group | F | P |
|---|---|---|---|---|---|---|
| Bacteroidetes | 45.07 ± 4.92 | 10.22 ± 2.32 | 43.54 ± 6.93 | 18.24 ± 16.89 | =16.04 | <0.001 |
| Proteobacteria | 27.67 ± 8.10 | 27.74 ± 14.28 | 28.31 ± 10.81 | 16.38 ± 6.08 | =2.06 | =0.14 |
| Fusobacteria | 16.81 ± 10.55 | 42.53 ± 8.38 | 19.42 ± 9.59 | 9.57 ± 6.56 | =11.40 | <0.001 |
| Firmicutes | 9.91 ± 5.45 | 18.71 ± 7.51 | 7.88 ± 3.04 | 46.54 ± 10.73 | =36.49 | <0.001 |
| Actinobacteria | 0.02 ± 0.01 | 0.07 ± 0.06 | 0.01 ± 0.01 | 8.35 ± 9.93 | =2.10 | =0.15 |

Notes:
The significances of Bacteroidetes, Firmicutes, Fusobacteria, and Proteobacteria were determined using the general linear model, whereas the generalized linear models was used to examine the significances of Actinobacteria.
Na, *Naja atra* group; Pmu, *Ptyas mucosus* group; Ec, *Elaphe carinata* group; Da, *Deinagkistrodon acutus* group.

(*Kundu et al., 2017*; *Rosshart et al., 2017*; *Hu et al., 2017*). In the present study, we provided the first comparative study of gut microbiotas among several species of farmed snakes in southern China, and revealed the factor driving variation that will be useful for understanding the relationship between gut microbiota and host species.

On average, we obtained 33,059 sequences per snake species (Table S1), consistent with previous similar studies. For example, a mean of 33,690 sequences were obtained in the forest musk deer (*Moschus berezovskii*) and the alpine musk deer (*Moschus chrysogaster*) (*Hu et al., 2017*); a mean of 30,000 sequences were obtained in crocodile lizards (*Shinisaurus crocodilurus*) (*Jiang et al., 2017*); and a mean of 16,307 sequences were obtained in black bears (*Song et al., 2017*). Thus, that the mean number of sequences and the depths of the sequencing data for each individual sample we obtained were reasonable (Table S1; Fig. S1).

## Dominant gut microbes

Bacteroidetes, Proteobacteria, Firmicutes, Fusobacteria, and Actinobacteria were the five most abundant phyla in the gut microbiota of the four farmed snake species (Fig. 1A). This differed from mammals (*Ley et al., 2008a*), birds (*Waite & Taylor, 2014*), and other reptiles (*Colston, Noonan & Jackson, 2015*; *Keenan, Engel & Elsey, 2013*; *McLaughlin, Cochran & Dowd, 2015*; *Jiang et al., 2017*). In previous studies of vertebrates, the gut microbiota have been dominated by the phyla Bacteroidetes and Firmicutes, which influence the physiological functions of the host with respect to metabolism and immunity (*Thomas et al., 2011*).

Lizards are another major taxon of reptiles (~60%) (*Uetz, Hošek & Hallermann, 2016*). Previous reports have indicated that the gut microbiota of lizards is dominated by the phyla Firmicutes (2.6–73%), Bacteroidetes (6.2–32.1%), and Proteobacteria (19.1–56.4%) (*Hong et al., 2011*; *Ren et al., 2016*; *Jiang et al., 2017*; *Kohl et al., 2017*). Proteobacteria enrichment in the human gut was an indicator of gut microbiota imbalance and was associated with host disease (*Shin, Whon & Bae, 2015*). However, the proportion of Proteobacteria in the gut microbiota of lizards was relatively high, although this proportion varied greatly by species. A similar situation has been reported in snakes. For example, the gut microbiota of the Burmese python (*Python bivittatus*) was

10.1% Proteobacteria (*Costello et al., 2010*), while that of the Timber rattlesnake (*Crotalus horridus*) was 85.0% Proteobacteria (*McLaughlin, Cochran & Dowd, 2015*). Similar results were also observed in the farmed snake species analyzed here (16.4–36.9%) (Table 2).

The proportion of Fusobacteria in the gut microbiotas of mammals, birds, and other snakes was relatively small (*Ley et al., 2008a*; *Costello et al., 2010*; *Waite & Taylor, 2014*; *Colston, Noonan & Jackson, 2015*; *McLaughlin, Cochran & Dowd, 2015*). However, Fusobacteria was a core gut microbiome of the American alligator (*Alligator mississippiensis*), which could affect lumen biofilm development (*Keenan, Engel & Elsey, 2013*). Here, Fusobacteria dominated the gut microbiotas of the farmed snakes; this is compositionally distinct from other vertebrate gut microbiomes, including those of other reptiles, fish, birds, and mammals.

*Bacteroides* and *Cetobacterium* were the dominant bacterial genera in gut microbiota of the farmed snakes (Fig. 2). *Bacteroides* maintain a complex and beneficial relationship in the host gut, and the symbiotic relationships between these bacteria and their hosts have been widely studied (*Thomas et al., 2011*). For example, *Bacteroides* species have complex systems for sensing nutrient utilization, regulating nutrient metabolism, and acquiring and hydrolyzing otherwise indigestible dietary polysaccharides (*Xu et al., 2003*). *Bacteroides* species control host gut homeostasis by interacting with the host immune system (*Wexler, 2007*). Here, the gut microbiotas of the farmed snakes were dominated by *Bacteroides*, especially the samples from *E. carinata* (42.09%) and *N. atra* (40.17%) (Fig. 3), indicating that the gut microbiota in snakes are species dependent. All *Cetobacterium* species are obligate anaerobes in phylum Fusobacteria (Fig. 2). *Cetobacterium* was the dominant genus in the gut microbiotas of all the farmed snakes analyzed herein; this is the first report of the dominance of this genus in the gut microbiotas of snakes.

## Fusobacteria in gut microbiotas of farmed snakes

Fusobacteria is a little-studied bacterial phylum, with a somewhat uncertain phylogenetic position (*Keenan, Engel & Elsey, 2013*). The results of the present study indicated that only two genera fell into Fusobacteria by phylogenetic analysis, *Cetobacterium*, and *Fusobacterium* (Fig. 2). However, it is possible that Fusobacteria includes additional unclassified genera, and/or that the Fusobacteria have been undersampled in previous studies of gut microbiotas (*Keenan, Engel & Elsey, 2013*). Previous studies have suggested that Fusobacteria have a core genome dissimilar to that of other bacterial lineages (*Mira et al., 2004*). Phylogenetic and comparative genomics analyses indicate that this phylum is closely affiliated with Bacteroidetes and Firmicutes, and may be derived from the Firmicutes (*Mira et al., 2004*). Phylogenetic analysis recovered a close relationship between Fusobacteria and Bacteroidetes, indicating a relatively close evolutionary relationship (Fig. 2). Bacteroidetes is one of the major lineages of bacteria, arising early in bacterial evolution (*Wexler, 2007*). Therefore, the evolutionary relationship between Fusobacteria and Bacteroidetes should be further investigated.

Fusobacteria species play a critical role in initial biofilm development (*Mira et al., 2004*), suggesting that the presence of these species in the guts of the farmed snakes

may affect the development of the lumen membrane (*Keenan, Engel & Elsey, 2013*). *Cetobacterium* was first isolated from the intestinal contents of a porpoise and from the mouth lesion of a minke whale (*Balaenoptera acutorostrata*) (*Foster et al., 1995*). Species in this genus transform peptones and carbohydrate into acetic acid (*Edwards, Logan & Gharbia, 2015*). Because Fusobacteria and *Cetobacterium* dominated the gut microbiotas of the farmed snakes, species in these taxa were likely commensal inhabitants of snake guts. It is therefore possible to speculate that, in snakes, Fusobacteria, and *Cetobacterium* play important roles in digestive organ development and in nutritional metabolism.

### The relationship between gut microbiota and host species

Many factors affect the vertebrate gut microbiotas, including host species, diet, and age (*Ley et al., 2008b*; *Waite & Taylor, 2014*; *Hu et al., 2017*; *Jiang et al., 2017*). The gut microbiota may also vary in different regions of the gut tract (*Ley et al., 2008b*; *Waite & Taylor, 2014*). Diet and host species influence the composition of the gut microbiota more than other factors (*Waite & Taylor, 2014*). The gut microbiota of the Burmese python was dominated by Firmicutes and Bacteroidetes (*Costello et al., 2010*), while the gut microbiota of the timber rattlesnake was uniquely dominated by Proteobacteria (*McLaughlin, Cochran & Dowd, 2015*). Bacteroidetes, Firmicutes, and Proteobacteria also dominated the gut microbiota of the cottonmouth snake (*Colston, Noonan & Jackson, 2015*). Therefore, the dominant bacterial phyla vary based on snake species. However, diet, age, habitat, and research method varied in previous studies of snake microbiotas, which possibly affected the distribution of bacterial species abundance at the phylum level. Here, the composition of gut microbiota was unique to each species of farmed snake. The four species shared similar breeding modes, but the composition and diversity of the gut microbiota were more similar within species and more different between species. This, suggested a relationship between the composition and diversity of the gut microbiota and the host species. However, the fecal samples of *E. carinata* originated from a different farm from the other three, which may have had an impact on the study results. The composition and diversity of the *E. carinata* gut microbiota differed from those of the other three species. For example, the community richness estimate for the fecal samples of *E. carinata* was significantly lower than that of *N. atra* and *D. acutus* (Fig. 3A). *E. carinata* had higher abundances of the dominant genera *Edwardsiella*, *Escherichia*, and *Plesiomonas* (Fig. S4). We therefore hypothesize that variations among farms may also be a factor contributing to the composition and community structures of host gut microbiotas. However, the community diversity estimate for the fecal samples of *E. carinata* did not differ significantly from *N. atra* (Fig. 3B). Bray–Curtis distance, PCA, and Heatmap vertical clustering showed that the gut microbiotas of *E. carinata* and *N. atra* were somewhat similar. Therefore, farm variation may not be an important factor altering the gut microbiotas of farmed snakes. In addition, the species studied here were similar with respect to diet, health, farmed environment, and age. This suggested that host species was probably the important factor shaping the microbiot.

## CONCLUSION

The compositions of the gut microbiotas of four farmed snake species in southern China were different to those of other snakes and vertebrates. The gut bacteria of these four species fell into 15 phyla, 18 classes, 22 orders, 35 families, and 58 genera. The five most abundant phyla were Bacteroidetes, Proteobacteria, Firmicutes, Fusobacteria, and Actinobacteria, while the five most abundant genera were *Bacteroides*, *Cetobacterium*, *Clostridium*, *Plesiomonas*, and *Paeniclostridium*. This was the first report that Fusobacteria and *Cetobacterium* dominated the gut microbiotas of snake species. Gut microbiotal diversity was highest in *D. acutus* and lowest in *E. carinata*. There were interspecific differences in gut microbiota composition and diversity among the four farmed snake species. Our results supported our hypothesis that host species was an important factor affecting the gut microbiotas of snakes. Further studies of snake gut microbiotas should investigate the relationship between phylogenetic position and function, as well as the characteristics of dominant bacteria that were unclassifiable. It is important to determine whether the immunity and growth of farmed snake populations can be improved by inoculating fecal suspensions generated by healthy wild snakes into the guts of farmed conspecifics.

### Funding

This study was funded by the National Natural Science Foundation of China (No. 31472021) and the Project for Wildlife Conservation and Management of the State Forestry Administration of China (No. 16180617). The funders had no role in the study design, data collection and analysis, decision to publish, or preparation of the manuscript.

### Grant Disclosures

The following grant information was disclosed by the authors:
National Natural Science Foundation of China: 31472021.
Wildlife Conservation and Management of the State Forestry Administration of China: 16180617.

### Competing Interests

Xinguo Gong is chairman of Qiyang Gong Xinguo Breeding Co., Ltd. The authors declare that they have no competing interests.

### Author Contributions

- Bing Zhang conceived and designed the experiments, performed the experiments, analyzed the data, contributed reagents/materials/analysis tools, wrote the paper, prepared figures and tables, authored or reviewed drafts of the paper.
- Jing Ren performed the experiments, analyzed the data, and wrote the paper.
- Daode Yang conceived and designed the experiments, authored or reviewed drafts of the paper, approved the final draft.

- Shuoran Liu conceived and designed the experiments, authored and reviewed drafts of the paper.
- Xinguo Gong performed the experiments.

## Animal Ethics

The following information was supplied relating to ethical approvals (i.e., approving body and any reference numbers):

This study was performed in accordance with the recommendations of the Institution of Animal Care and the Ethics Committee of Central South University of Forestry and Technology (approval number: CSUFT NS #20175167).

## Data Availability

Data is available at NCBI SRA: SRR8494339–SRR8494360.

The data is also available at figshare: Zhang, Bing; Ren, Jing; Yang, Daode; Liu, Shuoran; Gong, Xinguo (2019): The raw data of the article—Comparative analysis and characterization of the gut microbiota of four farmed snakes from southern China. figshare. Fileset. DOI 10.6084/m9.figshare.7769975.v1.

## Supplemental Information

Supplemental information for this article can be found online at http://dx.doi.org/10.7717/peerj.6658#supplemental-information.

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
