# Peer review of "Comparative analysis and characterization of the gut microbiota of four farmed snakes from southern China"

_PeerJ, doi:10.7717/peerj.6658_

## Round 0.1 · original submission · Major Revisions

All reviewers would like to see substantial revision to this ms, and they have all made good comments that you should follow in your revision. In particular, I would like to draw your attention to the following points.

Reviewers 1 & 2 both consider that the methods (in particular) are inadequate to replicate the study. Reviewer 1 has provided an annotated manuscript that details where improvements are required, but in general the authors need to re-write this manuscript and critically ask themselves whether or not it would be possible to repeat the study given the details they provide. If sufficient detail isn’t possible, the ms will be rejected.

This study would benefit greatly from a clear hypothesis at the outset.

It is very important to discuss the results with respect to the methods. You sampled snakes from two different farms, yet found that they were more similar to each other than snakes within the same farm. This must be discussed.

Lastly, I agree with all reviewers that the introduction and discussion should be expanded to include literature on other reptiles.

Reviewer 1 ·

Basic reporting

This English used in this manuscript is generally ok although I would encourage the authors to use active rather than passive voice.

The Introduction is not adequate and in several instances inappropriate references are used. I have indicated areas in need of improvement and suggestions for doing so directly in comments on the pdf.

Experimental design

The methods are insufficient to replicate the study. Specific details are included in the comments on the pdf but in brief:

Sampling design is not thoroughly explained.
Abundance data should be rarefied or justification provided why is was not.

Validity of the findings

I cannot adequately assess the validity of findings without more thorough explanation of methodology, but I have attempted to address specific concerns I have with comments directly on the pdf.

In general the statistical analyses performed should be adequately rigorous to address the authors' aims.

Additional comments

This is an extremely interesting paper and the authors should be commended for their effort. However I find some fatal flaws in the description of methodology, insufficient or inaccurate background information and overstated conclusions that prevent me from recommending the paper in its current form. However, with major revisions I hope to see this study resubmitted if the flaws in methodology can be addressed.

Annotated reviews are not available for download in order to protect the identity of reviewers who chose to remain anonymous.

Reviewer 2 ·

Basic reporting

The major objective of this study was to assess whether differences in gut microbiota composition of 4 farmed snake species correlated with the host species.

Despite the poor representation of reptiles, particularly snakes in the microbiota literature, this study is purely descriptive and shows a very superficial analysis of the microbiota composition, which fails to meet the high technical standards.

The raw data are not available.

Experimental design

Major flaw of this study are the lack of a clear hypothesis and the methods.
Methods are poorly described and show several inconsistencies, raising serious doubts about the reliability of the data and robustness of the analyses.

Below are some major concerns.

The Authors indicated that fecal “samples from the same species were pooled in the same centrifuge tube”, while the results of gut microbiota are shown for individual specimens.

Also, the Authors indicated the use of a paired-end approach for amplicon sequencing (line 95), but the results indicate an “average length of 252 bp” (line 141). Please clarify.

The analysis of the raw data is minimally described. How were the “paired-end reads” merged? How were they filtered? Were singletons removed? How were the sequences aligned for tree construction? All these important steps need to be indicated. The full pipeline used should be explained.

The confidence threshold of 0.60 for taxonomic assignment of OTUs is extremely low. The minimum acceptable threshold is usually 0.80 (line 104). Also, the RDP set 14 database is outdated.

The Authors indicate that a principal component analysis (PCA) was used, while they clearly performed a principal coordinate analysis (PcoA).

Dominant species were defined as those with relative abundance greater than 10% (line 114). This seems a personal choice and should be justified.

Line 147: “pooled sample”? Unclear, since individual gut microbiota composition is shown.

Validity of the findings

The main hypothesis, that is that host species is a major determinant in the structuring the gut microbiota of these four snakes species was not statistically tested. Results indicated “the PCA showed that the gut microbiotas from the same group were more similar to each other than to the gut microbiotas from different groups” (line 179-180). This is just a visual observation, and as shown in Figure S6 (which should go into the main manuscript) Ec and Da are indistinguishable. Also, giving that the host species sampled came from two distinct farms, factors associated to the distinct environments (despite a similar diet and habitat condition) can be important variables. This should be considered and discussed.

Among the several indexes used to estimate alpha diversity, which should be limited to no more than two to avoid redundancy, simpson’s diversity shows an opposite pattern with respect to all others. This is clearly an error in plotting, but surprisingly the Authors report these results without commenting.


Figure S1 shows a rank curve for evenness that is obscure. How can you assess evenness from this figure, if not sequencing depth is reported?



The Discussion is limited to a comparison of taxonomic composition of these four snakes species to published data from other vertebrates. Lizards, which are the reptiles with the most extensive microbial dataset currently available are not properly integrated into the discussion.

Additional comments

Overall, I suggest the Authors to perform a more thoroughly analysis of the data, displaying the details of the pipeline and testing with statistical analyses their major objective.

·

Basic reporting

In line 68, replace “southern China and teste that host species...” with “southern China and to evaluate whether host species...”

In the Introduction section, lines 49 - 58 the authors refer to gut Microbiome studies on birds and mammals. The inclusion of studies on reptiles (other than snakes), by virtue of their phylogenetic relationship, would potentially add greater value to the paper.

Experimental design

There is a need to perhaps include a statement relating to the feeding of the snakes (line 76) particularly in the context that: (i) one of the species was collected from a different site and (ii) that the regimen or proportions of feeding included multiple food sources (chicks and mice). The authors should indicate whether the feeding was entirely random in terms of these alternate sources and whether there is any indication of whether the same feeding patterns were applied at both sites of collection. This is with respect to the fact that the gut Microbiome composition in other spp. have been shown to be affected by different levels of protein and fat content, which would be the case for chicks and mice.

Line 85. The authors indicate that the individual samples were collected and then spooled according to species, followed by downstream freezing prior to DNA extraction. They should indicate that the individual samples were also treated in the same manner as the pooled samples, if indeed this was the case.

Line 99. Authors to indicate how the sequences were filtered and the low quality reads removed.

Lines 114 and 115. What was the basis for selecting the respective cutoff values. If taken from known studies, provide references.

Line 116. Is there an available references in support of the methodology employed.

In the section: “Comparisons of gut microbiotas among groups” starting at line 158, there is mention of analyses conducted in Qiime and Mothur. Do the authors have specific motivation for implementing different analyses using different packages.

Validity of the findings

Line 177. The observation of the lower diversity of gut microbiota in sample Ec could would be explored in the context that it is from a different site as the other three. The potential impact should be stated, in order to indicate that the authors are aware of it.

An evaluation of the number of reads generated for each of the individual samples indicate an average of 33059 reads (Table S1). What is the implication of this relatively low read number on the overall findings that were reported. How does this number compare to other studies of a similar nature. How does this number correlate with the overall diversity that was sampled versus the total available biodiversity (eg. Include a representation via a rarefaction curve).

Additional comments

no comment

---

## Round 0.2 · Minor Revisions

Thank you for your revision. This has been looked at by one reviewer and myself. We both agree that there are some outstanding minor issues that need to be addressed before your manuscript can be accepted.

1. Please check the specific name P. mucosus. The reptile database suggests that it should be P. mucosa: http://reptile-database.reptarium.cz/species?genus=Ptyas&species=mucosa

2. R version: I am concerned that the release date for v3.1.1 is July 2014. Is there a reason why a 2018 version was not used?

3. I am concerned by your choice to deposit raw data in a supplementary file. https://peerj.com/about/policies-and-procedures/ PeerJ policies suggest use of appropriate databases, and there are 2 such for this type of genetic material: GenBank and EMBL. Please reconsider where you deposit your raw data. (I would also encourage you to add a Supplementary file with your R scripts (as well as updating your R version and associated packages)).

4, In the discussion, you use the word "group" to refer to both species and groups of species. Please reconsider to aid readability.

5. In my previous decision, I drew attention to your use of snakes from different farms and your failure to consider the implications. You have now clarified this in terms of the M&M, but in L281, you do not use your results to discuss the findings in respect of the different farms (not sites) used. I think there is an important finding here that you are ignoring.

6. Some attention is required on the grammar (see highlights in attachment).

·

Basic reporting

No comment

Experimental design

The authors in their response to the initial review (point number 8, from reviewer 3) indicate that the use of different software for analyses does not affect results of the analyses. They however fail to indicate the basis for this statement. Could they provide other supporting material (references) supporting this.

Validity of the findings

The authors in their response to the initial review (point number 10, from reviewer 3) indicate that the number of reads evaluated in this study is similar to a number of other studies, which they mention. They, however, do not provide the references to these studies. Could they perhaps include this as part of their discussion, with the necessary references to support their work.

Additional comments

No comment

---

## Round 0.3 · accepted · Accept

Thank you for your revisions.

#